# Profiling malaria infection among under-five children in the Democratic Republic of Congo

**Jacques B. O. Emina** [1,2]*, **Henry V. Doctor**[3], **Yazoumé Yé**[4]

**1** Population and Health Research Institute, Kinshasa, Democratic Republic of Congo, **2** Department of Population and Development Study, University of Kinshasa, Kinshasa, Democratic Republic of Congo, **3** World Health Organization, Regional Office for the Eastern Mediterranean, Cairo, Egypt, **4** ICF, United States of America

* Jacques.emina@gmail.com

## Abstract

**Data Availability Statement:** This study is based on the Democratic Republic of Congo 2013 Demographic and Health Survey. These data are available at https://www.dhsprogram.com. Access to the file required authorization from The

### Introduction

In 2018, Malaria accounted for 38% of the overall morbidity and 36% of the overall mortality in the Democratic Republic of Congo (DRC). This study aimed to identify malaria socioeconomic predictors among children aged 6–59 months in DRC and to describe a socioeconomic profile of the most-at-risk children aged 6–59 months for malaria infection.

### Materials and methods

This study used data from the 2013 DRC Demographic and Health Survey. The sample included 8,547 children aged 6–59 months who were tested for malaria by microscopy. Malaria infection status, the dependent variable, is a dummy variable characterized as a positive or negative test. The independent variables were child's sex, age, and living arrangement; mother's education; household's socioeconomic variables; province of residence; and type of place of residence. Statistical analyses used the chi-square automatic interaction detector (CHAID) model and logistic regression.

### Results

Of the 8,547 children included in the sample, 25% had malaria infection. Four variables—child's age, mother's education, province, and wealth index—were statistically associated with the prevalence of malaria infection in bivariate analysis and multivariate analysis (CHAID and logistic regression). The prevalence of malaria infection increases with child's age and decreases significantly with mother's education and the household wealth index. These findings suggest that the prevalence of malaria infection is driven by interactions among environmental factors, socioeconomic characteristics, and probably differences in the implementation of malaria programs across the country. The effect of mother's education on malaria infection was only significant among under-five children living in Ituri, Kasaï-Central, Haut-Uele, Lomami, Nord-Ubangi, and Maniema provinces, and the effect of wealth index was significant in Mai-Ndombe, Tshopo, and Haut-Katanga provinces.

Demographic and Health Surveys Program (https://www.dhsprogram.com). The website depicts the process.

**Funding:** This research did not receive any specific funding. Yazoumé Yé (YY) is Vice President Malaria Surveillance and Research at ICF, USA, but was involved as a researcher. His institution did not have any role in the study design, data collection and analysis, decision to publish, or preparation of the manuscript. The specific roles of these authors are articulated in the 'author contributions' section.

**Competing interests:** Yazoumé Yé is an employee of M&E Infectious Disease, ICF International. This does not alter our adherence to PLOS ONE policies on sharing data and materials. There are no patents, products in development or marketed products associated with this research to declare. This does not alter our adherence to PLOS ONE policies on sharing data and materials.

**Abbreviations:** CHAID, chi-square automatic interaction detector; DHS, Demographic and Health Survey; DRC, Democratic Republic of Congo; ITN, insecticide-treated net; NMCP, national malaria control program; WHO, World Health Organization.

## Conclusion

Findings from this study could be used for targeting malaria interventions in DRC. Although malaria infection is common across the country, the prevalence of children at high risk for malaria infection varies by province and other background characteristics, including age, mother's education, wealth index, and place of residence. In light of these findings, designing provincial and multisectoral interventions could be an effective strategy to achieve zero malaria infection in DRC.

## Background

Malaria remains a threat to gains in health and development [1–5], even though the number of deaths due to this disease has been halved since 2000 [3, 4, 6]. One component of the Global Technical Strategy for Malaria 2016–2030 in the African region is to accelerate efforts toward elimination of malaria and attainment of malaria-free status by 2030 [7].

Malaria is a major public health concern in the Democratic Republic of Congo (DRC) [8], which accounts for 11% of the global malaria burden. In 1998, DRC created a national malarial control program (NMCP). The main objective of this program was to reduce malaria mortality by 50% and malaria morbidity by 25% by 2015. NMCP activities consist of the following interventions: distribution of insecticide-treated nets (ITNs), promotion of indoor residual spraying, promotion and implementation of intermittent preventive treatment in pregnancy, promotion of rapid diagnostic tests, and implementation of community and mother case management with artemisinin-based combination therapies [9, 10].

Over the past 15 years, the NMCP has intensified implementation of malaria control strategies [9]. The number of health zones covered by the NMCP has increased, from 271 out of 516 health zones in 2009 to all health zones (516) in 2016 [9]. In addition, the NMCP established a network of 11 sentinel sites for integrated surveillance of priority diseases, including malaria in 2003, with an expansion to 26 new provinces in 2016 [9].

Despite this progress, malaria accounted for 38% of overall morbidity and 36% of overall mortality in 2018 [4, 8–10] in the country. In 2018, the country reported 15 million malaria cases and about 27,458 malaria deaths [8]. A majority of malaria cases and rapid progression to death occur in young children [8–10]. Therefore, the gap between the current prevalence of malaria and the World Health Organization (WHO) goal of zero malaria case by 2030 remains important for DRC.

During the past three decades, interest in understanding factors associated with malaria among under-five children has increased in sub-Saharan Africa, but less is known about the socioeconomic profile of under-five children at higher risk of malaria infection in DRC. Published studies on malaria infection among under-five children can be divided into three groups. Some studies have analyzed trends in the prevalence of malaria infection among under-five children at the national and sub-national (province, district, and health zone) levels [11, 12]. Others have described socioeconomic factors associated with malaria morbidity and mortality among patients attending health facilities [13–15]. Some other studies have analyzed individual and community socioeconomic factors associated with malaria infection [16–18].

This situation is due to data limitations. The 2013 DRC Demographic and Health Survey (DHS) is the first national survey that collected data on malaria status, individual characteristics, maternal variables, and household characteristics. Using chi-square and regression models, previous studies did not automatically explore interactions between variables. The most-

at-risk groups for malaria infection could be a result of different interactions between malaria risk factors. For instance, existing literature revealed that children living in poorest households, children of less-educated mothers, children living in rural areas, and children aged above 23 months had greater risk of malaria infection [13–18]. However, a child could belong to all these risk categories, or could belong to the least risk groups and high risk groups at the same time [8].

Against this backdrop, this study aims to identify socioeconomic predictors of malaria and describe a socioeconomic profile of malaria prevalence among under-five children in DRC, who are considered to be one of most-at-risk groups affected by malaria, using chi-square, logistic regression, and chi-square automatic interaction detector (CHAID). Findings from this study will allow for the design of targeted interventions and evidence-based prevention programs as well as optimize coverage, reduce costs, and lower the number of new infections, given the high cost of interventions (194 million USD in 2013) [9, 10].

## Malaria risk factors and those most-at-risk for malaria

People are at risk of acquiring malaria infection due to factors related to environment, demographics, socioeconomic status, and exposure to prevention interventions [6, 11–18].

Fig 1 presents selected risk factors and categories under higher and lower risk of malaria using the WHO classification of malaria epidemiology [19].

Environmental and ecological factors, including distance from a household to the nearest body of water, altitude, temperature, and rainfall, determine malaria transmission zone: high transmission zone (mesoendemic, hyperendemic, and holoendemic) or low transmission zone (hypoendemic) [19, 20]. In this study, province of residence has been used as a proxy of geographical location.

Considering socioeconomic factors, children living in poor households, children whose mothers are less educated, and children living in rural areas are more likely to suffer from malaria [13–26]. Individuals belong to several categories, which in turn might belong to different malaria clusters (high risk or low risk).

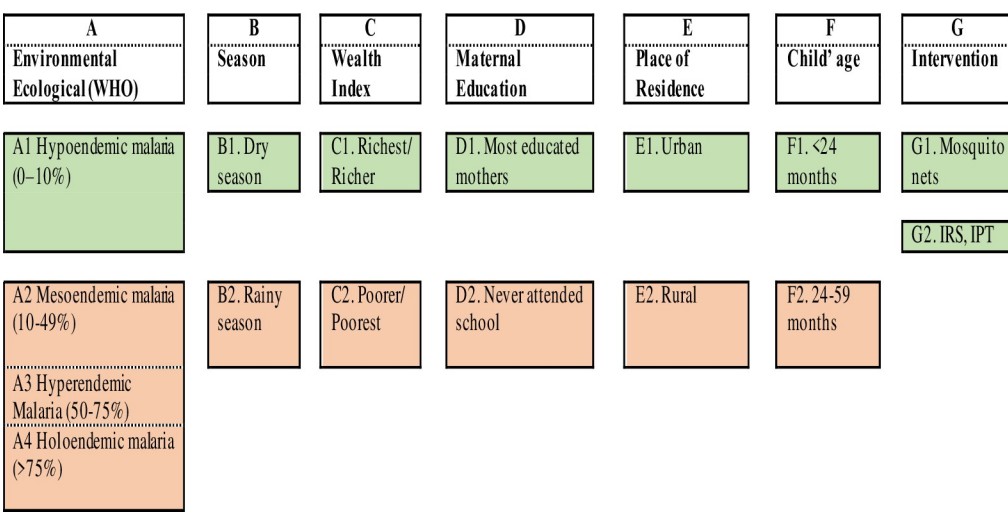

**Fig 1. Predictors of malaria among under five children and risk/vulnerable groups.** Source: Authors based on malaria literature.

## Data and methods

### Variables

The dependent variable for this analysis is the malaria infection status defined as a positive or negative malaria test. The independent variables include 10 variables grouped into 3 major types: (1) child variables (sex, age, living arrangement, whether slept under an ITN the night preceding data collection); (2) mother's education and household's characteristics (sex of the head of household, age of the head of household, wealth index); and (3) contextual factors, including province and place of residence. The choice of these variables is guided by the literature on factors associated with malaria infection [6, 13–26].

### Data sources

This study used data from the 2013 DRC DHS. The survey used a two-stage stratified-cluster sampling design based on the sampling frame of the 1984 Population and Housing Census, which was partially updated several times by administrative censuses and in the context of the presidential and legislative elections of 2011. The final survey unit chosen was the cluster (district or village), and, in total, 540 clusters were drawn. The first stage of sampling involved the selection of clusters known as primary sampling units. The second stage of sampling involved the selection of households from each cluster. Stratification in the first stage was achieved by grouping the 11 provinces into urban and rural areas. Primary sampling units in provinces with a very small population were selected with equal size allocation.

The DHS data offer a unique opportunity to profile malaria infection in the country due to the paucity of routine data, which are associated with unknown denominator and selection bias because all malaria cases are not reported in health facilities. In addition, those data do not include individual and household characteristics considered as predictors of the epidemy. The DHS incorporated five biomarker tests, including malaria testing. Malaria testing was carried out among children aged 6–59 months in half of the 18,360 selected households using microscopy. Using a finger (or heel) prick, a drop of blood was collected on a slide to prepare a thick film. All health technicians were trained to perform finger (or heel) pricks in the field according to the manufacturer's instructions. A total of 8,547 children aged 6–59 months were tested for malaria. The survey report provides more details on the sampling and microscopy process [26].

### Statistical analyses

Statistical analyses relied on Pearson's χ2, using the CHAID decision-tree algorithm implemented in SPSS V.21, and logistic regression. Pearson's χ2 was performed to identify associations between the malaria infection (positive or negative) and independent variables, including socioeconomic and demographic characteristics. The study applied the nominal CHAID model to identify the most significant determinants of malaria infection among under-five children and to describe the characteristics of the most-at-risk children for malaria infection considering interactions between predictors [27–31]. The model operates sequentially by recursively splitting under-five children into separate and distinct segments called nodes. The variation of the prevalence of malaria infection is minimized within each node and is maximized between nodes. After the initial splitting of the population (under-five children who received a malaria test) into different nodes based on the most significant predictor, the model repeats the process on each of the nodes until no significant predictors remain or until the number of observations in the node does not allow further partitions. Ideally the minimum

number of cases is estimated at 50 cases for child nodes, although the minimum number of cases can be lowered [27–30].

CHAID displays outcomes in hierarchical tree-structured form, in which the root is the population, which in this case is under-five children who received a malaria test. The root node, 'Node 0' or 'initial segment,' is the outcome variable, and subsequent levels include parent node and child node. Parent node is the upper node compared with nodes on the subsequent (lower) level, whereas any sub-node of a given node is called a child node. Sibling nodes are nodes on the same hierarchical level under the same parent node. Ancestor nodes comprise all nodes higher than a given node in the same lineage, and all nodes below the given node are called descendants. The terminal nodes are any node that does not have child nodes. They are the last categories of the CHAID tree. Findings include a table with five major columns describing each terminal node regarding content, population size, number with malaria infection, and the prevalence of malaria infection [27–31]. The analysis is focused on column 4, prevalence of malaria infection in each terminal node (category).

The study also employed logistic regression to identify predictors of malaria infection among under-five children. This consists of comparing the proportions using the logarithms of the odds ratio (log-odds). For each selected category, the model estimates the parameter ß (the ratio between the logit of a selected group and that of the reference group) and calculates the odds ratios while specifying their significance level (95% in this case) [31, 32]. If the odds ratio is equal to one, there is no difference between the considered group and the reference group regarding the risk of malaria infection. If the odds ratio is less than unity, children in the considered group are less likely to suffer from malaria infection, compared to children in the reference group. By contrast, if the odds ratio is greater than one, children in the selected group are more likely to suffer from malaria infection than children in the reference group [30–32]. However, the logistic regression model fails to incorporate non-monotonic relationships. Furthermore, it does not automatically detect interactions between segments or categories of independent variables. Significant differences have been established at $p < 0.05$.

## Data analysis strategies

We weighted data (CHAID in SPSS) and applied the SVY (logistic regression in STATA 15) to account for the complex design of the household survey. Missing values were treated as a separate category. For instance, the "Do not know" category for mother's education included children whose mother's education was missing.

## Ethical considerations

The DHS questionnaire, procedures, and testing protocol underwent a host country ethical review (by the DRC School of Public Health Ethical Review Committee) and were reviewed by ICF institutional review board. Participation in the individual survey and in malaria testing was voluntary, and parents signed the consent form before the interview and before their child's blood collection.

Interviews and biomarker testing were performed as privately as possible. Results of interviews and biomarker testing were strictly confidential. Only the DHS research team (interviewers, health specialists, editors, and supervisors) were allowed to access the data, essentially for communications. Each respondent's interview and biomarker data files were identified only by a series of numbers. The questionnaire cover sheets containing identifier numbers were destroyed after data processing.

## Results

### Participants

Table 1 shows the distribution of the study population by selected background characteristics. Of the total 8,547 children aged 6–59 months who were tested for malaria, 50% were female.

The distribution of the sample by age shows that 11% of the population was aged 6–11 months. The average age of the sample was estimated at 32.4 months (standard deviation 15.7). Children living with both parents constituted about 64% of the sample. A majority of participants lived in rural areas (71%) and in households headed by males (77%). Half of the participants (50%) were living in households headed by people aged 25–39 years, and 4% were living in households headed by people aged 65 years or above. By province, the sample size

**Table 1. Descriptive characteristics of children aged 6–59 months who had a malaria test, Democratic Republic of Congo Demographic and Health Survey, 2013.**

| Background variables | % | N | Background variables | % | N |
|---|---|---|---|---|---|
| **Sex of child** | | | **Province** | | |
| Male | 49.8 | 4,258 | Kinshasa | 5.3 | 453 |
| Female | 50.2 | 4,289 | Kwango | 4.4 | 379 |
| **Child's age group (months)** | | | Kwilu | 5.0 | 424 |
| 6–11 | 11.0 | 943 | Mai-Ndombe | 3.9 | 331 |
| 12–23 | 22.0 | 1,882 | Kongo Central | 4.5 | 386 |
| 24–35 | 22.2 | 1,894 | Equateur | 2.9 | 245 |
| 36–47 | 22.7 | 1,943 | Mongala | 3.5 | 299 |
| 48–59 | 22.1 | 1,885 | Nord-Ubangi | 3.0 | 255 |
| **Living arrangement** | | | Sud-Ubangi | 3.7 | 314 |
| Not living with mother | 9.0 | 770 | Tshuapa | 2.8 | 239 |
| Living with mother only | 26.5 | 2,266 | Kasaï | 4.4 | 374 |
| Living with both parents | 64.5 | 5,511 | Kasaï-Central | 4.3 | 368 |
| **Mother's education** | | | Kasaï-Oriental | 3.6 | 310 |
| None | 19.8 | 1,689 | Lomami | 4.7 | 405 |
| Primary | 40.8 | 3,490 | Sankuru | 3.5 | 299 |
| Secondary and above | 30.4 | 2,597 | Haut-Katanga | 3.5 | 295 |
| Don't know | 9.0 | 771 | Haut-Lomami | 3.5 | 295 |
| **Wealth index** | | | Lualaba | 2.5 | 209 |
| Poorest | 27.0 | 2,305 | Tanganyka | 3.3 | 279 |
| Poorer | 22.4 | 1,915 | Maniema | 4.9 | 415 |
| Middle | 20.2 | 1,730 | Nord-Kivu | 5.7 | 483 |
| Richer | 17.6 | 1,503 | Bas-Uele | 2.6 | 225 |
| Richest | 12.8 | 1,094 | Haut-Uele | 2.6 | 223 |
| **Sex of the head of household** | | | Ituri | 3.4 | 291 |
| Male | 77.3 | 6,605 | Tshopo | 3.1 | 266 |
| Female | 22.7 | 1,942 | Sud-Kivu | 5.7 | 485 |
| **Age of the head of household (years)** | | | **Place of residence** | | |
| <25 | 5.8 | 498 | Capital and large cities | 12.8 | 1,094 |
| 25–34 | 34.6 | 4,321 | Small cities and towns | 16.5 | 1,409 |
| 35–44 | 29.8 | 2,042 | Countryside | 70.7 | 6,044 |
| 45–64 | 25.6 | 948 | **Used ITN last night** | | |
| 65+ | 4.2 | 738 | No | 47.1 | 4,022 |
| | | | Yes | 52.9 | 4,525 |
| Total | **100.0** | **8,547** | | | |

varied, from almost 3% (Lualaba) to almost 6% (Sud-Kivu). About half of children tested (53%) slept under an ITN the night preceding the survey. Regarding the household wealth index, 27% of children were living in the poorest households, and 13% were living the richest households.

## Factors associated with malaria prevalence: Findings from the bivariate analysis

Overall, out of 8,547 children considered, 25% (95% confidence interval [CI]: 24.3%-26.2%) had malaria infection. Table 2 reports the prevalence of malaria infection among under-five children in DRC by selected background characteristics. Of the 10 independent variables included in the study, 6 were statistically significantly associated with malaria infection status.

**Table 2. Malaria prevalence by selected socioeconomic characteristics.**

| Variables | % | N | Chi-sq | P-value | Variables | % | N | Chi-sq | P-value |
|---|---|---|---|---|---|---|---|---|---|
| **Sex of child** | | | | | **Place of residence** | | | | |
| Male | 25.7 | 4,258 | 1.085 | 0.298 | Capital and large cities | 17.6 | 1,094 | | |
| Female | 24.7 | 4,289 | | | Small cities and towns | 24.6 | 1,409 | 41.177 | <0.001 |
| **Child's age group (months)** | | | | | Rural | 26.8 | 6,044 | | |
| 6–11 | 14.5 | 943 | | | **Province** | | | | |
| 12–23 | 19.1 | 1,882 | 159.688 | <0.001 | Kinshasa | 16.6 | 453 | | |
| 24–35 | 25.2 | 1,894 | | | Kwango | 9.0 | 379 | | |
| 36–47 | 30.4 | 1,943 | | | Kwilu | 8.0 | 424 | | |
| 48–59 | 31.4 | 1,885 | | | Mai-Ndombe | 29.3 | 331 | | |
| **Living arrangement** | | | | | Kongo Central | 26.4 | 386 | | |
| Not living with mother | 30.8 | 770 | | | Equateur | 17.6 | 245 | | |
| Living with mother only | 23.6 | 2,266 | 15.927 | <0.001 | Mongala | 15.4 | 299 | | |
| Living with both parents | 25.1 | 5,511 | | | Nord-Ubangi | 36.9 | 255 | | |
| **Mother's education** | | | | | Sud-Ubangi | 20.4 | 314 | | |
| None | 29.0 | 1,689 | | | Tshuapa | 15.9 | 239 | | |
| Primary | 28.3 | 3,490 | 135.171 | <0.001 | Kasaï | 30.0 | 374 | | |
| Secondary and above | 17.0 | 2,597 | | | Kasaï-Central | 38.0 | 368 | 611.068 | <0.001 |
| Don't know | 30.7 | 771 | | | Kasaï-Oriental | 27.7 | 310 | | |
| **Sex of the head of household** | | | | | Lomami | 38.3 | 405 | | |
| Male | 25.6 | 6,605 | 1.848 | 0.174 | Sankuru | 19.7 | 299 | | |
| Female | 24.1 | 1,942 | | | Haut-Katanga | 26.1 | 295 | | |
| **Age of the head of household (years)** | | | | | Haut-Lomami | 23.4 | 295 | | |
| <25 | 24.5 | 498 | | | Lualaba | 41.6 | 209 | | |
| 25–34 | 24.9 | 4,321 | | | Tanganyka | 50.2 | 279 | | |
| 35–44 | 23.8 | 2,042 | 8.880 | 0.064 | Maniema | 35.7 | 415 | | |
| 45–64 | 27.5 | 948 | | | Nord-Kivu | 8.1 | 483 | | |
| 65+ | 25.1 | 738 | | | Bas-Uele | 43.6 | 225 | | |
| **Wealth index** | | | | | Haut-Uele | 35.0 | 223 | | |
| Poorest | 26.9 | 2,305 | | | Ituri | 37.1 | 291 | | |
| Poorer | 29.7 | 1,915 | | | Tshopo | 26.7 | 266 | | |
| Middle | 27.2 | 1,730 | 132.722 | <0.001 | Sud-Kivu | 12.8 | 485 | | |
| Richer | 24.4 | 1,503 | | | **Used ITN last night** | | | | |
| Richest | 11.8 | 1,094 | | | No | 28.7 | 4,022 | 49.108 | <0.001 |
| **Total** | 25.2 | **8,547** | | | Yes | 22.1 | 4,525 | | |

Child's sex and sex and age of the head of household were not statistically associated with the likelihood of malaria infection.

The prevalence of malaria infection regularly increased with age. The percentage of children with malaria was estimated at 14% among children aged 6–11 months and 31% among those aged 48–59 months. The prevalence of malaria infection was low among children living with their mothers alone (23%) or living with both parents (25%), compared to children living with others (31%). Table 2 also shows a significant negative association between mother's education and malaria infection among under-five children: Malaria prevalence was higher among children whose mother did not attend school (29%) and lower among children whose mother had a secondary or higher education (17%).

The prevalence of malaria infection was higher in rural areas (27%) and small cities and towns (25%) than in large cities, including Kinshasa, the capital city (17%). The proportion of children with malaria was lower in the richest households (12%), compared to those living in all other households (from poorest to richer). Findings also show low prevalence of malaria infection (22%) among children who slept under an ITN the previous night, compared to those who did not sleep under an ITN (29%). The malaria endemicity shows regional heterogeneity, with a higher prevalence (50%) observed in Tanganyika province. Children living in Kwango (9%), Kwilu (8%), and Nord-Kivu (8%) had the lowest prevalence of malaria infection.

## Socioeconomic predictors of malaria: Findings from the CHAID model

Table 3 shows summary information on the specifications used to build the final CHAID model. Ten independent variables were examined, and five of those were statistically significant in the final model.

The CHAID tree diagram depicted in Fig 2A shows that the province of residence ($\chi2$ = 603.06, p<0.001) is the best predictor of malaria infection. Fig 2A–2E and Table 4 report predictors of malaria infection among under-five children in DRC by province.

Depending on province, the main predictors include child's age, wealth index, place of residence, and mother's education. No subsequent malaria infection predictor was identified in Tanganyika. In Haut-Lomami, Sankuru, and Sud-Ubangi, child's age is the only significant predictor of malaria prevalence ($\chi2$ = 19.61, p<0.001).

**Table 3. Malaria prevalence among under-five children: Summary of CHAID model.**

| Model components | Model specification | Results |
|---|---|---|
| Dependent variable | Parasitemia (via microscopy) in children aged 6–59 months | 25% |
| Independent variables | Child's sex, child's age, child's living arrangement, ITN used the night before the survey, place of residence, mother's education, age of the head of household, sex of the head of household, province, wealth index | Province, child's age, place of residence, mother's education, wealth index |
| Maximum tree depth | 3 | 3 |
| Minimum number of children in parent node | 100 | 100 |
| Minimum number of children in child node | 50 | 50 |
| Number of nodes | Na | 42 |
| Number of terminal nodes | Na | 26 |

In Bas-Uele and Lualaba, place of residence (χ2 = 12.57, p<0.001) is the only significant predictor of malaria infection among under-five children.

## Malaria infection among under-five children: Risk groups

The CHAID model splits participants into 26 homogeneous sub-groups, or terminal nodes, regarding the prevalence of malaria infection. Fig 2A–2E depict the process of creating the homogeneous groups, including the variables comprising each category. Table 5 describes these groups by their size (columns A and B), number of children with malaria infection (column C), the share in children with malaria infection (column E), and the proportionality of the share in malaria epidemic compared to the demographic weight (column F). The 26 homogenous sub-groups could be grouped into 4 major clusters (the third cluster includes

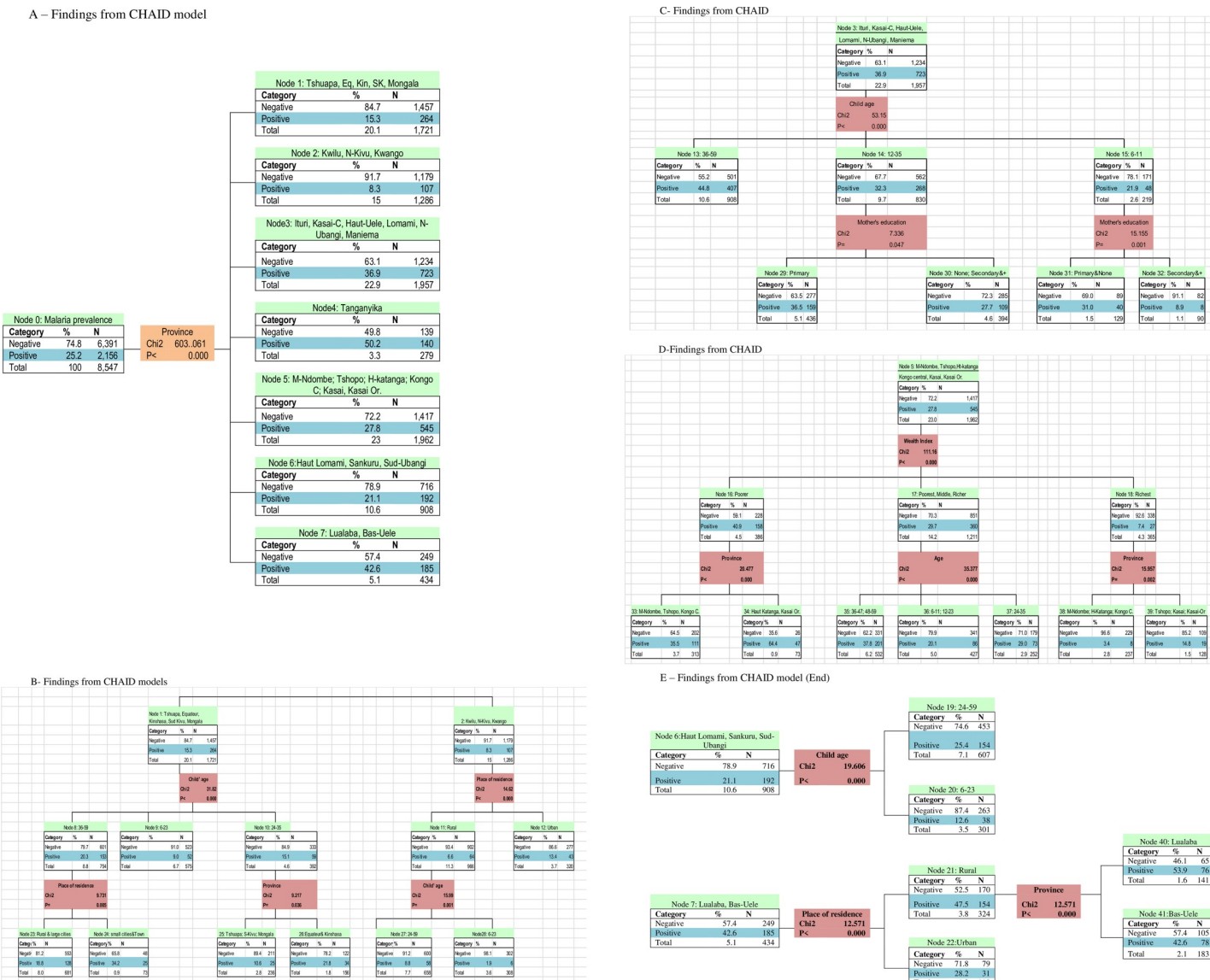

**Fig 2.** (A-E) Findings from CHAID model (End).

**Table 4. Socioeconomic predictors of malaria infection among under-five children by province, Democratic Republic of Congo Demographic and Health Survey, 2013.**

| Province | First predictor | Second predictor |
|---|---|---|
| Equateur, Kinshasa, Mongala, Sud-Kivu, Tshuapa | Child's age ($\chi2 = 31.82$, $p<0.001$) | Place of residence ($\chi2$ depends on child's age and province) |
| Kwango, Kwilu, Nord-Kivu | Place of residence ($\chi2 = 14.62$, $p<0.001$) | Child's age ($\chi2$ depends on place of residence and province) |
| Haut-Uele, Ituri, Kasaï-Central, Lomami, Maniema, Nord-Ubangi | Child's age ($\chi2 = 53.15$, $p<0.001$) | Mother's education ($\chi2$ depends on mother's education category and province) |
| Tanganyika | - | - |
| Haut-Katanga, Kongo Central, Kasaï, Kasaï-Oriental Mai-Ndombe, Tshopo, | Wealth index ($\chi2 = 111.16$, $p<0.001$) | Child's age ($\chi2$ depends on wealth index and province) |
| Haut-Lomami, Sankuru, Sud-Ubangi | Child's age ($\chi2 = 19.61$, $p<0.001$) | - |
| Bas-Uele, Lualaba | Place of residence ($\chi2 = 12.57$, $p<0.001$) | - |

two sub-clusters), consistent with WHO classification of malaria epidemiology [19]. Table 5 reports the characteristics of each group.

## Cluster 1—Children living in poor households in Haut-Katanga and Kasaï-Oriental

Children in this cluster represent 1% of participants and 2% of children who tested positive for malaria infection, yielding an index of 255%. Malaria prevalence was estimated at 64% in this cluster.

## Cluster 2—Children living in Tanganyika and rural Lualaba

The prevalence of malaria infection was estimated at 51.4% among children living in Tanganyika province and the rural area of Lualaba. This cluster accounts for 4% of participants and 10% of children with malaria infection, yielding an index of 204%. Like Cluster 1, it is located in the southern belt of DRC.

## Cluster 3—Mixed socioeconomic categories and different provinces

Cluster 3 includes the larger group of children (72.4% of children who received a malaria test). Malaria prevalence was estimated at 28.5%, ranging from 10.6% (children aged 24–35 months living in Tshuapa, Sud-Kivu, and Mongala) to 45% (children aged 36–59 months living in Ituri, Kasaï-Central, Haut-Uele, Lomami, Nord-Ubangi, and Maniema). Children in this cluster represent 82% of children with malaria infection. They live in 25 out of 26 provinces. This cluster includes children belonging to all age groups and socioeconomic characteristics (poorest-richest, living in rural and urban areas, children whose mothers never attended school, and children whose mothers had primary to secondary education).

## Cluster 4—Young children or living in high socioeconomic strata in 20 provinces

This cluster includes 22% of children who received a malaria test. The prevalence of malaria was estimated at 7% and accounts for 6% of all children with malaria. This cluster includes five subgroups: (1) children aged 6–23 months and living in Tshuapa, Equateur, Kinshasa, Sud-Kivu, and Mongala; (2) children aged 6–11 months and living Ituri, Kasaï-Central, Haut-Uele,

**Table 5. Chi-square automatic interaction detector risk groups.**

| Node description | Node size | | Tested + | | Prevalence | Index[f] |
|---|---|---|---|---|---|---|
| | N[a] | %[b] | N[c] | %[d] | (%)[e] | |
| *Cluster 1* | **493** | **5.8** | **263** | **12.2** | **53.3** | **211.5** |
| Poorer in Haut-Katanga and Kasaï-Oriental | 73 | 0.9 | 47 | 2.2 | 64.4 | 255.2 |
| Living in rural area in Lualaba | 141 | 1.6 | 76 | 3.5 | 53.9 | 213.7 |
| Living in Tanganyika | 279 | 3.3 | 140 | 6.5 | 50.2 | 198.9 |
| *Cluster 2* | **6,186** | **72.4** | **1,761** | **81.7** | **28.5** | **112.9** |
| Age 36–59 months; living in Ituri, Kasaï-Central, Haut-Uele, Lomami, Nord-Ubangi, Maniema | 908 | 10.6 | 407 | 18.9 | 44.8 | 177.7 |
| Living in rural area in Bas-Uele | 183 | 2.1 | 78 | 3.6 | 42.6 | 169.0 |
| Age 36–59 months; in poorest/middle/richer quintiles; living in Mai-Ndombe, Tshopo, Haut-Katanga, Kongo Central, Kasaï, Kasaï-Oriental | 532 | 6.2 | 201 | 9.3 | 37.8 | 149.8 |
| Age 12–35 months; mother with primary education; living in Ituri, Kasaï-Central, Haut-Uele, Lomami, Nord-Ubangi, Maniema | 436 | 5.1 | 159 | 7.4 | 36.5 | 144.6 |
| In poorer quintile; living in Mai-Ndombe, Tshopo, Kongo Central, Kasaï | 313 | 3.7 | 111 | 5.1 | 35.5 | 140.6 |
| Age 36–59 months; living in small cities/towns in Tshuapa, Equateur, Sud-Kivu, Mongala | 73 | 0.9 | 25 | 1.2 | 34.2 | 135.8 |
| Age 6–11 months; living in Ituri, Kasaï-Central, Haut-Uele, Lomami, Nord-Ubangi, Maniema; mother with none or primary education | 129 | 1.5 | 40 | 1.9 | 31.0 | 122.9 |
| Age 24–35 months; in poorest/middle/richer quintiles; living in Mai-Ndombe, Tshopo, Haut-Katanga, Kongo Central, Kasaï, Kasaï-Oriental | 252 | 2.9 | 73 | 3.4 | 29.0 | 114.8 |
| Living in small cities/towns (Bas-Uele, Lualaba) | 110 | 1.3 | 31 | 1.4 | 28.2 | 111.7 |
| Age 12–35 months; mother with none or secondary and above education; living in Ituri, Kasaï-Central, Haut-Uele, Lomami, Nord-Ubangi, Maniema | 394 | 4.6 | 109 | 5.1 | 27.7 | 109.7 |
| Aged 24–59 months; living in Haut-Lomami, Sankuru, Sud-Ubangi | 607 | 7.1 | 154 | 7.1 | 25.4 | 100.6 |
| Aged 24–35 months; living in Kinshasa, Equateur | 156 | 1.8 | 34 | 1.6 | 21.8 | 86.4 |
| Age 6–23 months; in poorest/middle/richer quintiles; living in Mai-Ndombe, Tshopo, Haut-Katanga, Kongo Central, Kasaï, Kasaï-Oriental | 427 | 5.0 | 86 | 4.0 | 20.1 | 79.8 |
| Age 36–59 months; living in capital/large cities or rural areas in Tshuapa, Equateur, Kinshasa, Sud-Kivu, Mongala | 681 | 8.0 | 128 | 5.9 | 18.8 | 74.5 |
| In richest quintile; living in Tshopo, Kasaï, Kasaï-Oriental | 128 | 1.5 | 19 | 0.9 | 14.8 | 58.8 |
| Living in large cities/small cities and towns in Kwilu, Nord-Kivu, Kwango | 320 | 3.7 | 43 | 2.0 | 13.4 | 53.3 |
| Age 6–23 months; living in Haut-Lomami, Sankuru, Sud-Ubangi | 301 | 3.5 | 38 | 1.8 | 12.6 | 50.0 |
| Age 24–35 months; living in Tshuapa, Sud-Kivu, Mongala | 236 | 2.8 | 25 | 1.2 | 10.6 | 42.0 |
| *Cluster 3* | **1,868** | **21.9** | **132** | **6.1** | **7.1** | **28.0** |
| Age 6–23 months; living in Tshuapa, Equateur, Kinshasa, Sud-Kivu, Mongala | 575 | 6.7 | 52 | 2.4 | 9.0 | 35.9 |
| Age 6–11 months; living in Ituri, Kasaï-Central, Haut-Uele, Lomami, Nord-Ubangi, Maniema; mother with secondary and above education, mother's education unknown | 90 | 1.1 | 8 | 0.4 | 8.9 | 35.2 |
| Age 24–59 months; living in rural areas in Kwilu, Nord-Kivu, Kwango | 658 | 7.7 | 58 | 2.7 | 8.8 | 34.9 |
| In richest quintile; living in Mai-Ndombe, Haut-Katanga, Kongo Central | 237 | 2.8 | 8 | 0.4 | 3.4 | 13.4 |
| Age 6–23 months; living in rural areas in Kwilu, Nord-Kivu, Kwango | 308 | 3.6 | 6 | 0.3 | 1.9 | 7.7 |
| *Overall* | *8547* | *100.0* | *2156* | *100.0* | *25.2* | *100.0* |

Notes: N = number of children.

[a] Number of children who received malaria test.

[b] Demographic size in percentage = ([a] /Σ[a]) × 100.

[c] Number of children tested positive.

[d] Demographic size in percentage among children tested positive = ([c] /Σ[c]) × 100.

[e] Prevalence of malaria in each group = ([c] /Σ[a]) × 100.

[f] Node index (proportionality index) = [([c] /Σ[c]) /([a] /Σ[a])] × 100.

Lomami, Nord-Ubangi, and Maniema, whose mothers have secondary education and above or whose mother's education is unknown; (3) children aged 24–59 months and living in rural areas in Kwilu, Nord-Kivu, and Kwango; (4) children living in the richest quintile in Mai-Ndombe, Haut-Katanga, and Kongo Central; and (5) children aged 6–23 months and living in rural areas in Kwilu, Nord-Kivu, and Kwango.

## Socioeconomic predictors of malaria: Findings from the logistic regression model

Of the nine variables included in the logistic regression model (Table 6), five are statistically associated with the prevalence of malaria infection among under-five children: child's age, sleeping under an ITN, mother's education, household wealth index, and province of residence. The risk of malaria infection increases significantly with child's age. Compared to children aged 6–11 months, those aged 12–23 months have 1.7 times more risk of malaria (95% CI = 1.24–2.36). This risk is estimated at 2.6 (95% CI = 2.02–3.54) for children aged 24–35 months, 3.3 (95% CI = 2.53–4.46) for children aged 36–47 months, and 3.4 (95% CI = 2.53–4.62) for children aged 48–59 months.

The likelihood of malaria infection is low among under-five children living in the least poor households. The risk of malaria infection is 31% lower (odds ratio = 0.61; 95% CI = 0.50–0.97) among children living in richer households, and about 81% lower (odds ratio = 0.19; 95% CI = 0.09–0.37) among children living in the richest households, compared to children living in the poorest households. Considering mother's education, children whose mothers have secondary education have about 33% lower risk (odds ratio = 0.67; 95% CI = 0.51–0.87) of malaria infection, compared to those whose mothers did not attend school. There is no significant difference in the prevalence of malaria infection between children whose mothers attended only primary school and those whose mothers did not attend school. Children who slept under an ITN have 14% lower risk (odds ratio = 0.86; 95% CI = 0.74–0.99) of malaria infection, compared to children who did not sleep under an ITN.

After controlling for other variables, the risk of malaria infection among under-five children is lower in all provinces compared to Kinshasa. However, the difference is statistically significant in the following 10 provinces only: Kwango, Kwilu, Equateur, Mongala, Sud-Ubangi, Tshuapa, Sankuru, Tshopo, and Nord-Kivu (p-value: <0.05).

## Discussion

This study aimed to identify predictors of malaria infection among under-five children in DRC and describe the socioeconomic profile of children with malaria infection. The discussion is organized around three points: complexity of findings, complementarity between methodological approaches, and policy implications. Table 7 summarizes key findings and reports those that are consistent with the literature.

Of the 10 variables analyzed, 4 were statistically associated with the prevalence of malaria infection in bivariate analysis and multivariate analysis (CHAID and logistic regression): child's age, mother's education, province, and wealth index. These findings are consistent with previous studies [8, 9, 13–25]. The risk of malaria infection among under-five children increases with child's age. Two hypotheses, which we were not able to test in this study, may explain this finding. First, younger children may be protected from malaria because of the antibodies they acquire from their mother during pregnancy and during breastfeeding [33]. Second, younger children in some countries in sub-Saharan Africa, including DRC, share a bed with their mother and are more likely to be covered properly with a blanket or an ITN than

**Table 6. Factors associated with malaria infection among under-five children in the Democratic Republic of Congo: Findings from logistic regression.**

| | Odds ratio | P-value | 95% confidence interval | |
|---|---|---|---|---|
| *Sex of child* | | | | |
| Male | Reference | | | |
| Female | 0.962 | 0.578 | 0.841 | 1.101 |
| *Child's age group (months)* | | | | |
| 6–11 | Reference | | | |
| 12–23 | 1.713 | <0.001 | 1.245 | 2.357 |
| 24–35 | 2.675 | <0.001 | 2.020 | 3.543 |
| 36–47 | 3.356 | <0.001 | 2.526 | 4.459 |
| 48–59 | 3.417 | <0.001 | 2.527 | 4.622 |
| *Mother's education* | | | | |
| No education | Reference | | | |
| Primary | 0.965 | 0.773 | 0.754 | 1.233 |
| Secondary and above | 0.669 | 0.003 | 0.513 | 0.874 |
| Don't know | 0.892 | 0.447 | 0.665 | 1.198 |
| *Used ITN last night* | | | | |
| No | Reference | | | |
| Yes | 0.857 | 0.044 | 0.738 | 0.996 |
| *Sex of the head of household* | | | | |
| Male | Reference | | | |
| Female | 0.841 | 0.083 | 0.692 | 1.023 |
| *Age of the head of household (years)* | | | | |
| <25 | Reference | | | |
| 25–34 | 0.878 | 0.394 | 0.652 | 1.184 |
| 35–44 | 0.871 | 0.392 | 0.633 | 1.196 |
| 45–64 | 0.980 | 0.897 | 0.719 | 1.335 |
| 65+ | 0.779 | 0.248 | 0.510 | 1.190 |
| *Household wealth index* | | | | |
| Poorest | Reference | | | |
| Poorer | 1.201 | 0.120 | 0.953 | 1.512 |
| Middle | 1.000 | 0.997 | 0.765 | 1.309 |
| Richer | 0.692 | 0.028 | 0.498 | 0.962 |
| Richest | 0.187 | <0.001 | 0.095 | 0.369 |
| *Type of place of residence* | | | | |
| Capital, large city | Reference | | | |
| Small cities and towns | 1.084 | 0.810 | 0.562 | 2.090 |
| Rural | 0.808 | 0.510 | 0.429 | 1.524 |
| *Province of residence* | | | | |
| Kinshasa | Reference | | | |
| Kwango | 0.106 | <0.001 | 0.035 | 0.318 |
| Kwilu | 0.096 | <0.001 | 0.035 | 0.268 |
| Mai-Ndombe | 0.399 | 0.081 | 0.142 | 1.120 |
| Kongo Central | 0.447 | 0.084 | 0.179 | 1.114 |
| Equateur | 0.189 | 0.039 | 0.039 | 0.923 |
| Mongala | 0.256 | 0.015 | 0.085 | 0.769 |
| Nord-Ubangi | 0.464 | 0.119 | 0.176 | 1.218 |
| Sud-Ubangi | 0.211 | <0.001 | 0.082 | 0.542 |
| Tshuapa | 0.190 | <0.001 | 0.069 | 0.524 |

(*Continued*)

**Table 6.** (Continued)

|  | Odds ratio | P-value | 95% confidence interval | |
|---|---|---|---|---|
| Kasaï | 0.339 | 0.080 | 0.101 | 1.137 |
| Kasaï-Central | 0.634 | 0.319 | 0.259 | 1.555 |
| Kasaï-Oriental | 0.493 | 0.069 | 0.230 | 1.056 |
| Lomami | 0.614 | 0.299 | 0.245 | 1.543 |
| Sankuru | 0.175 | 0.002 | 0.057 | 0.532 |
| Haut-Katanga | 0.633 | 0.319 | 0.257 | 1.558 |
| Haut-Lomami | 0.394 | 0.094 | 0.132 | 1.174 |
| Lualaba | 0.837 | 0.764 | 0.261 | 2.681 |
| Tanganyka | 0.915 | 0.865 | 0.329 | 2.543 |
| Maniema | 0.441 | 0.087 | 0.173 | 1.128 |
| Nord-Kivu | 0.053 | <0.001 | 0.018 | 0.157 |
| Bas-Uele | 1.028 | 0.953 | 0.407 | 2.598 |
| Haut-Uele | 0.760 | 0.635 | 0.244 | 2.369 |
| Ituri | 0.512 | 0.199 | 0.184 | 1.423 |
| Tshopo | 0.344 | 0.029 | 0.132 | 0.897 |
| Sud-Kivu | 0.092 | <0.001 | 0.033 | 0.258 |
| _cons | 0.68647 | 0.377 | 0.2973776 | 1.584656 |

older children [34–36]. Fig 3 shows the proportion of under-five children who slept under an ITN the night preceding the study by age in DRC.

Findings also show that the higher the level of a mother's education, the lower the prevalence of malaria among under-five children. Previous studies reported that mothers with higher levels of education were more knowledgeable about malaria prevention and signs and were therefore more proactive and reactive regarding prevention than mothers with lower levels of education [34, 37–40]. In 2018, the proportion of under-five children who slept under an ITN was higher (more than 60%) among children of the most educated mothers (secondary education or higher), compared to children whose mothers did not reach that level of education (36% for children whose mothers did not attend school and 46% for children whose mothers attended only primary school) [39].

The results of this study also show that malaria cases were less prevalent among children from the richest households, compared to children from the poorest households. People in a higher wealth quintile are more likely to live in improved houses and more likely to be educated and have better access to knowledge about the steps to prevent malaria infection. They are also more likely to be able to afford ITNs and to use them correctly, as well as to be able to afford insecticides used for indoor spraying [5, 20, 38–40]. Data from the DRC 2018 Multiple

**Table 7. Summary of key findings.**

| Variables | Chi-square (bivariate) | Logistic regression | CHAID | Consistency with literature |
|---|---|---|---|---|
| Child's sex | Not significant | Not significant | Not significant | Yes |
| Child's age | Significant | Significant | Significant | Yes |
| ITN used the night before the survey | Significant | Significant | Not significant | Depends on method |
| Place of residence | Significant | Not significant | Significant | Depends on method |
| Mother's education | Significant | Significant | Significant | Yes |
| Province | Significant | Significant | Significant | Yes |
| Wealth index | Significant | Significant | Significant | Yes |

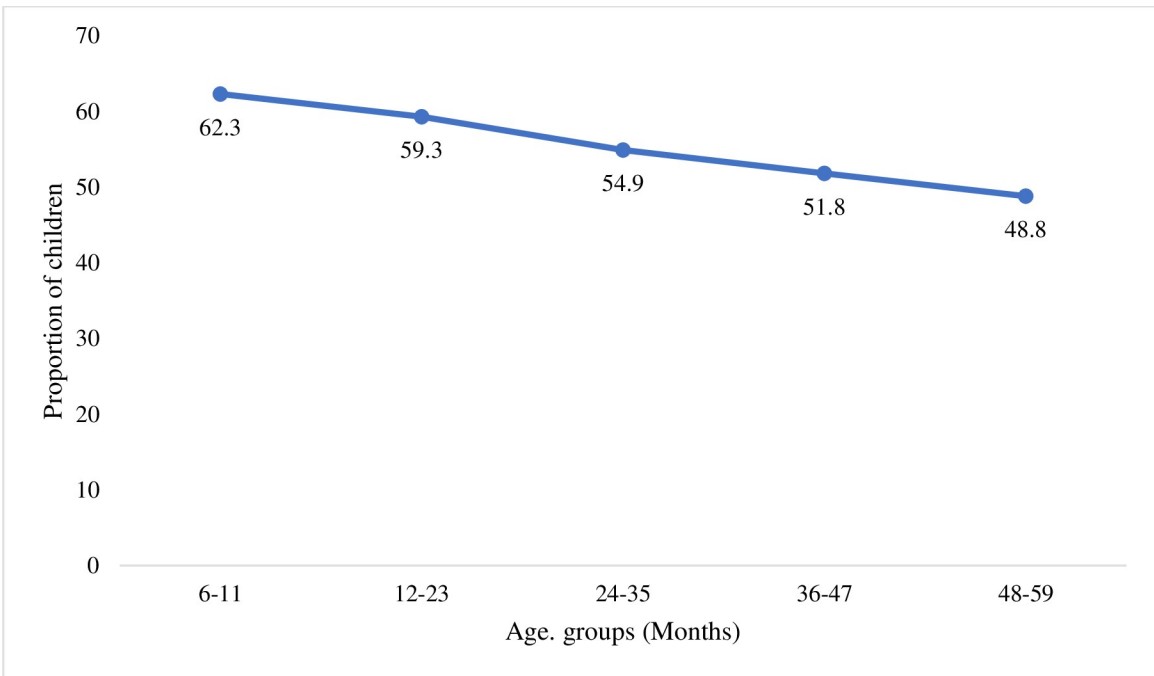

**Fig 3. Proportion of children who slept under a mosquito net the night preceding the study in DRC.**

Indicator Cluster Survey showed that the proportion of under-five children who slept under an ITN varied, from 35% in the poorest households to 69% in the richest households [39].

Provincial differences in malaria prevalence among under-five children could be explained by environmental and ecological factors, including distance from a household to the nearest body of water, altitude, temperature, and rainfall [6, 8, 9, 19, 20]. Data from the 2013 DHS as well as the 2018 MICS report variations in the use of ITNs among under-five children by province in DRC [26, 39].

Surprisingly, although bivariate analysis and logistic regression models report low prevalence of malaria infection among children who slept under an ITN (odds ratio = 0.86; 95% CI = 0.74–0.99; p-value = 0.04), compared to those who were not protected, the variable "slept under mosquito net" is not statistically associated with the prevalence of malaria if one considers the findings from the CHAID model. It is likely that this effect has been captured by other variables, such child's age, mother's education, household's wealth index, place of residence, and province of residence, which are associated with the use of ITNs by under-five children and with the prevalence of malaria among under-five children. In a previous study, Ferrari [16] found that the effect of mosquito nets was not significant in the lower transmission strata in DRC. That study reported that children aged less than two years were more likely to sleep under a mosquito net, compared to older children [16].

This study also shows that the effect of place of residence was not statistically significant in the logistic regression model. The CHAID model reported significant differences between children living in urban areas and those living in rural contexts, particularly among children living in: (1) Kwango, Nord-Kivu, Kwango, Lualaba, and Bas-Uele; and (2) Equateur, Mongala, Tshuapa, Sud-Kivu, and Kinshasa and aged 36–59 months. This difference could be explained by the fact that logistic regression does not automatically detect interactions between independent variables or the segments in which the model is statistically significant.

Comparison of findings by method of analysis also shows that the province of residence (spatial location) is the most important predictor of malaria prevalence among under-five children in DRC [8, 9, 13–20, 37]. The effect of other variables, such as child's age, place of residence, mother's education, or wealth index, depends on the province. For instance, mother's education is only significant for children aged 6–11 months and 12–35 months living in Ituri, Kasaï-Central, Haut-Uele, Lomami, Nord-Ubangi, and Maniema provinces. The findings also show that children belonging to the same socioeconomic category (e.g., age, place of residence, wealth index) might belong to different risk groups (high or less), depending on their region (province) of residence. These findings support results from Ferrari [16], which revealed that predictors of child malaria varied by strata (malaria high transmission zone versus malaria low transmission zone). Furthermore, these findings suggest that the prevalence of malaria infection is driven by interaction among environmental factors, socioeconomic characteristics, and probably differences in the implementation of malaria programs across the country. Table 8 summarizes key findings and recommendations.

Transforming the current malaria program to a "National Multisectoral Malaria Program" involving the Ministries of Health, Agriculture, Education, Urbanization and Habitat, Rural Development, Social and Humanitarian Affairs, Interior Affairs, Gender and Family, and Environment (Fig 4) will strengthen the fight again malaria. This institution should also involve key stakeholders working on malaria and other related programs, including members of civil society organizations, nongovernmental organizations, and academia.

The current national program should play the role of Technical Secretariat. Such an institution will be consistent with the United Nations Development Programme (UNDP) and

**Table 8. Summary of key findings and recommendations.**

| Key findings | Recommendations |
|---|---|
| Prevalence of malaria infection is driven by interactions between environmental factors and socioeconomic characteristics. | Rename the malaria program as the "National Multisectoral Malaria Program" involving the Ministries of Health, Agriculture, Education, Urbanization and Habitat, Rural Development, Social and Humanitarian Affairs, Interior Affairs, Gender and Family, and Environment (Fig 4). |
| High-risk groups for malaria exist in the majority of provinces. | Implement universal coverage of ITNs and house improvement in all provinces because they are the most cost-efficient intervention to reduce both burden and transmission, irrespective of the ecology within a setting [9, 18, 20]. Include malaria education in school curriculum as part of the stand-alone "Family Life Course" implemented in DRC schools because malaria is a public health problem with a prevalence of 25% among under-five children [41]. Integrate malaria prevention and care into workplace policies and use campaigns to raise awareness among employees [41]. |
| Spatial variation of malaria: malaria prevalence is high in some provinces. | Promote province-based implementation studies on malaria as well as malaria interventions. In high endemic clusters (prevalence above 40%), ITNs could be associated with seasonal malaria chemoprevention or indoor residual spraying [43–45]. |
| There is a high prevalence of malaria among children aged 24 months and above. | Increase vitamin A and zinc supplementation among under-five children as part of immunization [37]. |
| There is a high prevalence of malaria among children of mothers with low education and/or living in the poorest households and/or living in rural areas. | Promote the community engagement strategy for malaria prevention and treatment and share malaria prevention information on social media [41]. |

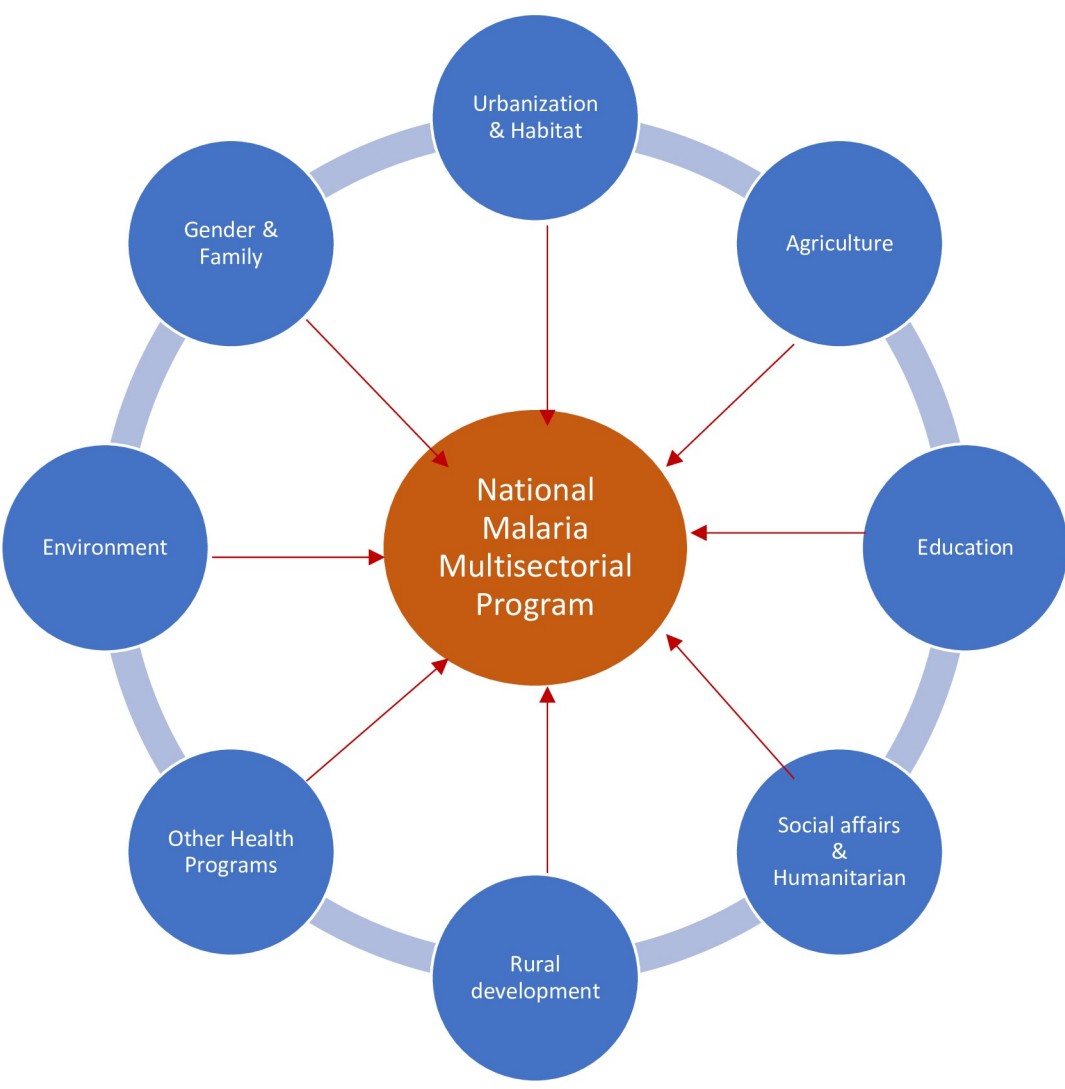

**Fig 4. Multisectoral malaria program.**

Malaria Roll Back (MRB) multisectoral framework for malaria [5]. It will design the national malaria policy and advise the government.

## Study limitations

This study has two methodological limitations, which do not affect its quality. First, the CHAID model does not consider the hierarchical structure of the DHS data, which might influence the overall prevalence of malaria infection. However, CHAID does allow automatic detection of segments in which the prevalence of malaria infection is similar and addresses the failure to incorporate non-monotonic relationships in logistic regression. Furthermore, CHAID is a diagnostic technique for partitioning the data set into several segments [31]. Because of the heterogeneity of the data, segment-wise prediction models (CHAID) are more advantageous than the logistic global model.

Second, the study did not control for the seasonality of malaria transmission. DRC is crossed by the equator, with rainy and dry seasons varying across the country by province and

by district and health zone within the same province. DHS data did not collect information about the season.

Missing values were treated as a separate category. For instance, the "Do not know" category for mother's education included children whose mother's education was missing.

## Conclusion

In summary, findings from this study could be used for designing malaria target interventions in DRC. They show heterogeneity in malaria burden among under-five children in DRC. Consistent with findings from previous studies, four of the nine variables included in multivariate models (logistic regression and CHAID) were statistically associated with the prevalence of malaria infection: child's age, mother's education, province, and wealth index. Furthermore, findings from the CHAID model reveal that predictors of malaria infection vary by province. In each province, child's age is the common predictor of malaria infection. Other predictors include place of residence, wealth index, and mother's education. These findings also suggest that the prevalence of malaria infection is driven by interactions among environmental factors, socioeconomic characteristics, and probably differences in the implementation of malaria programs across the country. The most-at-risk groups for malaria in one province might not be the ones at greater risk in other provinces. Therefore, designing provincial and multisectoral interventions could be the most effective strategies to achieve zero malaria infection in DRC. Some of the key interventions are outlined in the *World Malaria Report 2019* [42] and include investments in malaria programs and research, malaria prevention, diagnostic testing and treatment, and malaria surveillance systems.

Due to the multiplicity of factors that are linked to malaria transmission, it is important that various actions that directly and indirectly affect malaria prevention policy are designed and operate in synergy, through a multi-sectorial malaria prevention policy and national program.

## Acknowledgments

The authors thank DHS Programs for providing free access to the 2013 datasets for the Demographic and Health Survey. They also thank Michelle Tsagli for her comments on the earlier version of the paper.

## Author Contributions

**Conceptualization:** Jacques B. O. Emina, Henry V. Doctor.

**Data curation:** Jacques B. O. Emina.

**Formal analysis:** Jacques B. O. Emina.

**Methodology:** Jacques B. O. Emina, Henry V. Doctor, Yazoumé Yé.

**Validation:** Jacques B. O. Emina, Henry V. Doctor, Yazoumé Yé.

**Visualization:** Jacques B. O. Emina.

**Writing – original draft:** Jacques B. O. Emina, Henry V. Doctor, Yazoumé Yé.

**Writing – review & editing:** Jacques B. O. Emina, Henry V. Doctor, Yazoumé Yé.

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
