## [Decision Letter · Decision Letter 0]

4 Aug 2020

PONE-D-20-10152

Profiling malaria infection among under-five children in the Democratic Republic of Congo

PLOS ONE

Dear Dr. Emina,

Thank you for submitting your manuscript to PLOS ONE. After careful consideration, we feel that it has merit but does not fully meet PLOS ONE’s publication criteria as it currently stands. Therefore, we invite you to submit a revised version of the manuscript that addresses the points raised during the review process.

In particular, you should carefully consider all the points made by Reviewer 2.

We look forward to receiving your revised manuscript.

Kind regards,

Thomas A. Smith

Academic Editor

PLOS ONE

2. In Tables 2 and 3, please report more specific p-values.

3. In your ethics statement in the Methods section and in the online submission form, please provide additional information about the data used in your retrospective study. Specifically, please ensure that you have discussed whether all data were fully anonymized before you accessed them and/or whether the IRB or ethics committee waived the requirement for informed consent. Please see https://journals.plos.org/plosone/s/submission-guidelines#loc-personal-data-from-third-party-sources for our submission guidelines on this topic.

5. Please ensure that you include a title page within your main document. You should list all authors and all affiliations as per our author instructions and clearly indicate the corresponding author.

6. Please amend your manuscript to include your abstract after the title page.

7. Your ethics statement must appear in the Methods section of your manuscript. If your ethics statement is written in any section besides the Methods, please move it to the Methods section and delete it from any other section. Please also ensure that your ethics statement is included in your manuscript, as the ethics section of your online submission will not be published alongside your manuscript.

8. Please include your tables as part of your main manuscript and remove the individual files. Please note that supplementary tables (should remain/ be uploaded) as separate "supporting information" files

Reviewers' comments:

Reviewer's Responses to Questions

**Comments to the Author**

1. Is the manuscript technically sound, and do the data support the conclusions?

Reviewer #1: Yes

Reviewer #2: Partly

2. Has the statistical analysis been performed appropriately and rigorously? 

Reviewer #1: Yes

Reviewer #2: I Don't Know

3. Have the authors made all data underlying the findings in their manuscript fully available?

Reviewer #1: Yes

Reviewer #2: Yes

4. Is the manuscript presented in an intelligible fashion and written in standard English?

Reviewer #1: Yes

Reviewer #2: No

5. Review Comments to the Author

Reviewer #1: The study by Jacques and others, entitled “Profiling malaria infection among under-five children in the Democratic Republic of Congo”, aimed at identifying the malaria socioeconomic association and predictors among children aged 6-59 months and to provide socioeconomic data on the malaria most-at-risk children using Pearson’s chai square and Chi square Automatic Interaction Detector statistical model.

While the study is not novel in term of using this type of analysis, it is considered new for the malaria profile in these parts of Congo.

The authors need to go through the comments bellow:

1. Table 1 (descriptive characteristics):

• Age of child (months): to be change into Age group (months).

• Place of residence is categorized in the tables as 1) Capital & large cities and 2) Small cities & countryside, while it is Urban and Rural in text and figures.

Preferably change into urban and rural for consistency.

2. Table 2 (malaria prevalence by selected socioeconomic characteristics):

• Table is better to be presented in 2X2 format.

• Age group (months) as in table 1.

• Mother’s education:

change Secondary &+ into Secondary and above as in table 1 for consistency.

change DKN into Don’t know as in table 1 for consistency.

3. Table 3 (socioeconomic predictors):

• Chi square value in Node 1 province (Tshuapa, Equateur, Kinshasa, Sud Kivu, Mongala) for the child’s age was 14.62 while it was 31.82 in the analysis (figure 2B).

Please correct it.

4. Although the CHAID is a good analysis to find out the variable interaction associations, it seems not able to determine which group among the variable is responsible for the significant association (causal association between dependent and independent variable).

From my own opinion, it will be clear if the author can include a table of multivariate logistic regression for the significant variables as comparison.

Reviewer #2: Comments

This is an interesting paper looking at malaria predictors in DRC. The use of DHS data provides an opportunity to have a nation-wide data set and look at risk factors at national level.

However, the overall language of the manuscript needs improvement. Some of the recommendations and conclusions are not supported by the results presented. References need attention. There is a lot of unnecessary citations and important statements without references. The choice of words is not always standard. There are several statement that are either not accurate or not appropriate to malaria situation in DRC.

Specific comments

- Abstract: The abstract on the manuscript summary (editorial manager page 1) is different from the actual abstract on the manuscript document (manuscript page 1)

- Include results of measure of association in the abstract to support results

- I would remove the first sentence of the abstract.

- Line 9: increased from 271 in 2009 to 516 in 2016: out of how many? Could also specified covered with what? Malaria Interventions?

- Line 10: malaria accounted for 38% of global morbidity: the word global here is a bit confusing. Is it in DRC or Globally?

- Line 10: Remove advanced

- Line 17: The sample includes: included

- Line 17: malaria test: specify which one/ones?

- Line 19: positive or negative parasitemia test: I’d remove parasitemia

- Line 21: (2) mother’s and household’ socio-economic variables: education household’s characteristics (sex of the head of household, age of the head of household, wealth index) : which one is for mother?, check punctuation, maybe a comma missing after education

- Line 23: (3) contextual factors (province of residence and place of residence): what is meant by place of residence (is it rural vs urban?, if yes please specify)

- Line 26: (95% confidence interval 24.3%-26.2%): 95%CI and remove % inside brackets

- Lines 28-30: While predictors…..all provinces: check punctuation

- Line 31: effect of … on malaria infection, not prevalence

- Line 44: DRC is not a malaria eliminating country

- Background and methods sections: don’t have line numbers, making it difficult to reference comments

- Background: need some rewriting, some of the points are not relevant. Some unnecessary referencing, and references missing at important points

- Paragraph 1, line 3: Malaria epidemic…..: Not sure the author refer to an actual malaria epidemic or to the overall global malaria situation. I would delete “epidemic”

- Paragraph 1, line 4: Accounting for 11%...: reference

- Paragraph 1, line 6: An estimated 97%...: reference

- Paragraph 1, line 8: xxxx reported malaria deaths: delete reported

- Paragraph 2: first sentence: not sure about its relevance. WHO ….have developed”: I would replace develop by recommend.

- Despite this progress, malaria accounted for 38% of global morbidity…: is it at global level or national level? Seems like national level. Use of Global is a bit confusing. Maybe use overall

- Given the high cost of interventions, 194 million USD in 2013: reference

- Despite the growing literature…… especially in DRC: would be good to mention what has been done previously and the findings. I know for sure that some work has been done in DRC on malaria infection predictors.

- Against this backdrop, ….describe socioeconomic profile of malaria prevalence…by malaria: Profile of malaria infection

- Figure 1 presents …….endemicity class: check punctuation and grammar

- Figure 1: not clear if this is made by the author or just a general figure on malaria predictors. If the former, please make it clear and explain how these were selected. If the latter, please provide a reference. The reference provided is a very old reference on malaria stratification, this figure is not from there. As it is, the figure is hard to understand and is misleading. The color code is hard to understand. A given factor can be a predictor of malaria in different level of malaria endemicity.

- Hypoendemic malaria if the prevalence varies between 0 and 10%, Mesoendemic malaria if the prevalence comprises between 10 and 50%, Hyperendemic malaria if the prevalence varies between 50 and 75%, Holoendemic malaria if the malaria prevalence is estimated above 75%: I don’t think this is needed in the paper, as long as there’s a reference

- Individuals belong to several categories, which in their turn might belong…: remove “their”, not sure what the authors mean by categories. Do they mean risk groups?

- Some individuals might belong to categories included in the low malaria risk clusters: same as above about categories

- Last two paragraphs of “Malaria risk factors and vulnerable children for malaria”: there is a lot of unnecessary referencing. Additionally, I don’t see the relevance of mentioning all possible malaria interventions.

- Exposure to these interventions might likely explain socioeconomic differences in malaria prevalence among children aged less than five years [9, 18-48].:

o Related to the previous comment on unnecessary referencing, there are more than 30 references for this sentence only.

o I am not sure about this statement: The difference in socioeconomic status is more likely to explain difference in access to interventions.

- The dependent variable for this analysis is the malaria infection status characterized as positive or negative parasitemia test.: replace “characterized” by “defined” and “parasitemia” by “malaria”. The primary outcome is malaria infection. Authors should be consistent when referring to it, not change it to malaria prevalence or malaria epidemy.

- The data offer a unique …..the epidemy:

o The footnote can be integrated in the main text.

o Epidemy: as said above, be consistent when referring to your primary outcome

- Malaria testing was carried out among children aged 6-59 months in half of selected households. In the DHS, malaria test relied on microscopy (reading of thick-smear slide): Suggestion: Malaria testing using microscopy was carried out among children aged 6-59 months in half of selected households.

- The survey report provides more details on the sampling and microscopy process: the paper should present at least a summary of the sampling and selection of participants.

- Table 1 shows the distribution of the study population by selected background characteristics: All these results (participants characteristics) should go under results.

- Missing values were treated as a separate category. For instance, “Do not know” category for maternal education included children whose mother’s education was missing: This should go under analyses

- Statistical analyses:

o Too much details on CHAID model. Should summarize and provide references for more details

o Provide more information about how the CHAID model was constructed in this specific study. How were the explanatory variables selected and included? What is the level of significance?

o How was wealth index estimated?

o DHS is a cluster survey: was clustering taken into account? Please specify

o Was any survey weight applied?

o Consider carefully the choice of words (malaria structure?): do the authors mean malaria status?

- Study limitations:

o should be discussed in the discussion section.

o Please explain how the hierarchical nature of DHS data would affect the overall prevalence

- Results

o Participants characteristics (table 1) presented in “data sources” should be briefly presented here

o Line 3: the 95% CI is very narrow, suggesting that clustering was not taken into account

o Line 12: (23%-25%): what is this range for?

o Line 38: Box 1 and CHAID outputs show 5 significant predictors not 6 as mention here.

o Line 48: Missing residence

o Line 49: In Tanganyika, there is no predictor of malaria since the prevalence is very high (50%) for all children. Suggestion: No subsequent malaria infection predictor was identified in Tanganyika.

o Line 58: there’s no table 4

o Line 63: WHO classification: please specify classification of what?

o Clusters: Since these clusters are built by authors using the 26 final nodes of CHAID, I would expect malaria prevalence in each cluster to be presented as a range

o If the main purpose of creating the clusters is to summarize the population at risk (and protected) based on identified risk factors ( or protective factors), then cluster 3 is not informative as it includes the majority of the sample and all explanatory variables.

- Discussion

o Overall, the discussion is very limited and brief, to say the least. There are several points that could be discussed, but the authors shortly discussed 1 or 2 points only.

o The recommendations provided are not supported by the findings of this paper

o Line 97: demographic as well

o Line 103 and 104: predictor of malaria infection

o Line 104: These are not “the main predictor” but identified predictors of malaria infection

o Line 105: The sentence starting with

6. PLOS authors have the option to publish the peer review history of their article (what does this mean?). If published, this will include your full peer review and any attached files.

Reviewer #1: **Yes: **Wahib M. Atroosh

Reviewer #2: No

---

## [Author Response · Author response to Decision Letter 0]

25 Feb 2021

RESPONSES TO COMMENTS

1. In Tables 2 and 3, please report more specific p-values.

Answer: P-values have been inserted in Tables 2 and 3. Chi-square’ p-values are reported per variable, not per category of each variable.

3. In your ethics statement in the Methods section and in the online submission form, please provide additional information about the data used in your retrospective study. Specifically, please ensure that you have discussed whether all data were fully anonymized before you accessed them and/or whether the IRB or ethics committee waived the requirement for informed consent. Please see https://journals.plos.org/plosone/s/submission-guidelines#loc-personal-data-from-third-party-sources for our submission guidelines on this topic.

Answer: We have inserted the ethical considerations in the Data and Methods section by stating that The DHS questionnaire and the testing protocol undergo a host country ethical review as well as an ethical review at ICF Macro. Furthermore, participation in individual survey and in malaria testing is voluntary. Parents should sign the consent form before interview and before children’ blood collection.

Answer: The ORCID iD has been provided

5. Please ensure that you include a title page within your main document. You should list all authors and all affiliations as per our author instructions and clearly indicate the corresponding author.

Answer: Thank you for the comment which has been implemented.

6. Please amend your manuscript to include your abstract after the title page.

Answer: Thank you for the comment which has been implemented.

7. Your ethics statement must appear in the Methods section of your manuscript. If your ethics statement is written in any section besides the Methods, please move it to the Methods section and delete it from any other section. Please also ensure that your ethics statement is included in your manuscript, as the ethics section of your online submission will not be published alongside your manuscript.

Answer: We have included the ethical considerations in the Data and methods section.

8. Please include your tables as part of your main manuscript and remove the individual files. Please note that supplementary tables (should remain/ be uploaded) as separate "supporting information" files

Answer: Thank you for the comment which has been implemented. 

Reviewers' comments:

Reviewer's Responses to Questions

Comments to the Author

1. Is the manuscript technically sound, and do the data support the conclusions?

Reviewer #1: Yes

Reviewer #2: Partly

2. Has the statistical analysis been performed appropriately and rigorously? 

Reviewer #1: Yes

Reviewer #2: I Don't Know

3. Have the authors made all data underlying the findings in their manuscript fully available?

Reviewer #1: Yes

Reviewer #2: Yes

4. Is the manuscript presented in an intelligible fashion and written in standard English?

Reviewer #1: Yes

Reviewer #2: No

5. Review Comments to the Author

Reviewer #1: The study by Jacques and others, entitled “Profiling malaria infection among under-five children in the Democratic Republic of Congo”, aimed at identifying the malaria socioeconomic association and predictors among children aged 6-59 months and to provide socioeconomic data on the malaria most-at-risk children using Pearson’s chai square and Chi square Automatic Interaction Detector statistical model.

While the study is not novel in term of using this type of analysis, it is considered new for the malaria profile in these parts of Congo.

The authors need to go through the comments bellow:

1. Table 1 (descriptive characteristics):

• Age of child (months): to be change into Age group (months).

Answer: Thank you for the comment which has been implemented.

• Place of residence is categorized in the tables as 1) Capital & large cities and 2) Small cities & countryside, while it is Urban and Rural in text and figures.

Preferably change into urban and rural for consistency.

Answer: Thank you for the comment. We prefer to have three categories because small cities and town might be different to large cities and capital regarding socioeconomic infrastructures. However, sometime children belonging to the two sub-groups had similarities. In that case we can combine into the one group (urban)..

2. Table 2 (malaria prevalence by selected socioeconomic characteristics):

• Table is better to be presented in 2X2 format.

Answer: Thank you for the comment. We think that the current format is readable and avoid confusion.

• Age group (months) as in table 1.

Answer: Thank you for the comment which has been implemented.

• Mother’s education:

change Secondary &+ into Secondary and above as in table 1 for consistency.

change DKN into Don’t know as in table 1 for consistency.

Answer: Thank you for the comment which has been implemented.

3. Table 3 (socioeconomic predictors):

• Chi square value in Node 1 province (Tshuapa, Equateur, Kinshasa, Sud Kivu, Mongala) for the child’s age was 14.62 while it was 31.82 in the analysis (figure 2B).

Please correct it.

Answer: Thank you for the comment which has been implemented.

4. Although the CHAID is a good analysis to find out the variable interaction associations, it seems not able to determine which group among the variable is responsible for the significant association (causal association between dependent and independent variable).

Answer: Box 1 reports the CHAID model (variables included in the model and variables significantly associated with malaria infection among children aged less than five years. Significant differences have been established at p<0.05.

From my own opinion, it will be clear if the author can include a table of multivariate logistic regression for the significant variables as comparison.

Answer: Thank you for your comment. We have included the logistic regression model.

Reviewer #2: Comments

This is an interesting paper looking at malaria predictors in DRC. The use of DHS data provides an opportunity to have a nation-wide data set and look at risk factors at national level.

However, the overall language of the manuscript needs improvement. Some of the recommendations and conclusions are not supported by the results presented. References need attention. There is a lot of unnecessary citations and important statements without references. The choice of words is not always standard. There are several statement that are either not accurate or not appropriate to malaria situation in DRC.

Specific comments

- Abstract: The abstract on the manuscript summary (editorial manager page 1) is different from the actual abstract on the manuscript document (manuscript page 1)

Answer: Thank you for your comment which we have implemented.

- Include results of measure of association in the abstract to support results

Answer: Thank you for your comment. We have included the summary of findings from different methods. 

- I would remove the first sentence of the abstract.

Answer: Thank you for this comment which has been implemented.

“Though the number of health zones covered by the national malaria control program (PNLP) increased from 271 out 516 in 2009 to 516 in 2016, malaria accounted for 38% of global morbidity and 36% of overall mortality”.

- Line 9: increased from 271 in 2009 to 516 in 2016: out of how many? Could also specified covered with what? Malaria Interventions?

Answer: Thank you for this comment which has been implemented.

- Line 10: malaria accounted for 38% of global morbidity: the word global here is a bit confusing. Is it in DRC or Globally?

Answer: Thank you for this comment. We specified that:

Though the number of health zones covered by the national malaria control program (PNLP) increased from 271 out 516 in 2009 to 516 in 2016, malaria accounted for 38% of overall morbidity and 36% of overall mortality in DRC.

- Line 10: Remove advanced

Answer: Thank you for this comment which has been implemented.

- Line 17: The sample includes: included

Answer: Thank you for this comment which has been implemented.

- Line 17: malaria test: specify which one/ones?

Answer: Thank you for this comment. We have specified the malaria test performed: 

The sample included 8,547 children aged 6-59 months who got a malaria microscopy test (reading of thick-smear slide) among which half were female and 71% were living in rural areas.

- Line 19: positive or negative parasitemia test: I’d remove parasitemia

Answer: Thank you for this comment which has been implemented.

- Line 21: (2) mother’s and household’ socio-economic variables: education household’s characteristics (sex of the head of household, age of the head of household, wealth index) : which one is for mother?, check punctuation, maybe a comma missing after education.

Answer: Thank you for this comment. We have revised the sentence: 

“mother’s education and household’ socio-economic variables (sex of the head of household, age of the head of household, wealth index).

- Line 23: (3) contextual factors (province of residence and place of residence): what is meant by place of residence (is it rural vs urban?, if yes please specify)

Answer: Thank you for this comment which has been implemented as below:

“contextual factors (province of residence and type of place of residence (rural or urban))”.

- Line 26: (95% confidence interval 24.3%-26.2%): 95%CI and remove % inside brackets

Answer: Thank you for this comment which has been implemented.

- Lines 28-30: While predictors…..all provinces: check punctuation

Answer: Thank you for this comment. We have revised the sentence as follow:

“While predictors of malaria infection (type of place of residence, mothers’ education and wealth index) varied by province of residence, the effect of child’s age on malaria infection was significant in all provinces”.

- Line 31: effect of … on malaria infection, not prevalence

Answer: Thank you for this comment which has been implemented.

- Line 44: DRC is not a malaria eliminating country

Answer: We have changed the sentence.

- Background and methods sections: don’t have line numbers, making it difficult to reference comments

Answer: Thank you for this comment. We have added line numbers in these sections. 

- Background: need some rewriting, some of the points are not relevant. Some unnecessary referencing, and references missing at important points

- Paragraph 1, line 3: Malaria epidemic…..: Not sure the author refer to an actual malaria epidemic or to the overall global malaria situation. I would delete “epidemic”

Answer: Thank you for this comment which has been implemented.

- Paragraph 1, line 4: Accounting for 11%...: reference

Answer: Thank you for this comment. We have inserted the reference.

- Paragraph 1, line 6: An estimated 97%...: reference

Answer: Thank you for this comment. We have inserted the reference.

- Paragraph 1, line 8: xxxx reported malaria deaths: delete reported

Answer: Thank you for this comment which has been implemented.

- Paragraph 2: first sentence: not sure about its relevance. WHO ….have developed”: I would replace develop by recommend.

Answer: Thank you for this comment which has been implemented.

- Despite this progress, malaria accounted for 38% of global morbidity…: is it at global level or national level? Seems like national level. Use of Global is a bit confusing. Maybe use overall

Answer: Thank you for this comment. We have revised this sentence.

“Despite this progress, malaria accounted for 38% of overall morbidity and 36% of overall mortality [4,10] in DRC”.

- Given the high cost of interventions, 194 million USD in 2013: reference

Answer: Thank you for this comment. We have inserted the reference.

- Despite the growing literature…… especially in DRC: would be good to mention what has been done previously and the findings. I know for sure that some work has been done in DRC on malaria infection predictors.

- Against this backdrop, ….describe socioeconomic profile of malaria prevalence…by malaria: Profile of malaria infection

Answer: We have mentioned some existing publications on malaria in DRC. 

- Figure 1 presents …….endemicity class: check punctuation and grammar

Answer: Thank you for this comment. We have revised the sentence

- Figure 1: not clear if this is made by the author or just a general figure on malaria predictors. If the former, please make it clear and explain how these were selected. If the latter, please provide a reference. The reference provided is a very old reference on malaria stratification, this figure is not from there. As it is, the figure is hard to understand and is misleading. The color code is hard to understand. A given factor can be a predictor of malaria in different level of malaria endemicity.

Answer: Thank you for this comment. We have inserted source of Figure 1: Developed by Authors based on malaria literature.

- Hypoendemic malaria if the prevalence varies between 0 and 10%, Mesoendemic malaria if the prevalence comprises between 10 and 50%, Hyperendemic malaria if the prevalence varies between 50 and 75%, Holoendemic malaria if the malaria prevalence is estimated above 75%: I don’t think this is needed in the paper, as long as there’s a reference.

Answer: We have deleted the categories and just left the reference.

- Individuals belong to several categories, which in their turn might belong…: remove “their”, not sure what the authors mean by categories. Do they mean risk groups?

Answer: Thank you for this comment which has been implemented.

- Some individuals might belong to categories included in the low malaria risk clusters: same as above about categories.

Answer: We have removed this sentence since it was tautological.

- Last two paragraphs of “Malaria risk factors and vulnerable children for malaria”: there is a lot of unnecessary referencing. Additionally, I don’t see the relevance of mentioning all possible malaria interventions.

Answer: We have revised the section.

- Exposure to these interventions might likely explain socioeconomic differences in malaria prevalence among children aged less than five years [9, 18-48].:

Answer: We have dropped the paragraph on interventions.

o Related to the previous comment on unnecessary referencing, there are more than 30 references for this sentence only.

Answer: We have reduced number of references.

o I am not sure about this statement: The difference in socioeconomic status is more likely to explain difference in access to interventions.

Answer: We have revised the sentence.

- The dependent variable for this analysis is the malaria infection status characterized as positive or negative parasitemia test.: replace “characterized” by “defined” and “parasitemia” by “malaria”. The primary outcome is malaria infection. Authors should be consistent when referring to it, not change it to malaria prevalence or malaria epidemy.

Answer: We have edited accordingly based on the suggestions above.

- The data offer a unique …..the epidemy:

o The footnote can be integrated in the main text.

o Epidemy: as said above, be consistent when referring to your primary outcome

- Malaria testing was carried out among children aged 6-59 months in half of selected households. In the DHS, malaria test relied on microscopy (reading of thick-smear slide): Suggestion: Malaria testing using microscopy was carried out among children aged 6-59 months in half of selected households.

Answer: We have edited based on the suggestions above.

- The survey report provides more details on the sampling and microscopy process: the paper should present at least a summary of the sampling and selection of participants.

Answer: Thank you for this comment. We have inserted a paragraph summarizing the sampling process.

Malaria testing was carried out among children aged 6-59 months in half of the 18,360 selected households. Malaria testing using microscopy was carried out among children aged 6-59 months in half of selected households. Using a finger (or heel) prick, a drop of blood was collected on a slide to prepare a thick drop. After drying, the slides were stored in special boxes with desiccants and humidity controllers. These dishes were transferred regularly to the NRL for the detection of hematozoa by microscopy, which was carried out regularly. All health technicians were trained to perform finger (or heel) prick in the field according to the manufacturers’ instructions. A total of 8,547 children aged 6-59 months got malaria. The survey report provides more details on the sampling and microscopy process

- Table 1 shows the distribution of the study population by selected background characteristics: All these results (participants characteristics) should go under results.

Answer: Thank you for this comment which has been implemented.

- Missing values were treated as a separate category. For instance, “Do not know” category for maternal education included children whose mother’s education was missing: This should go under analyses

Answer: Thank you for this comment which has been implemented.

- Statistical analyses:

o Too much details on CHAID model. Should summarize and provide references for more details

Answer: We have summarized the description of CHAID model.

o Provide more information about how the CHAID model was constructed in this specific study. How were the explanatory variables selected and included? What is the level of significance?

Answer: Thank you for the comment. The Malaria risk factors and the most-at-risk for malaria section guides the selection of variables included in the study.

o How was wealth index estimated?

Answer: DHS datasets provide variable wealth index (hv270).

o DHS is a cluster survey: was clustering taken into account? Please specify

Answer: We weighted analyses to consider clustering.

o Was any survey weight applied?

Answer: The survey weight was applied. Proportions (%) are from the weighted sample while numbers are unweighted.

o Consider carefully the choice of words (malaria structure?): do the authors mean malaria status?

- Study limitations:

o should be discussed in the discussion section.

Answer: Thank you for the comment. We have displaced the study limitations into the discussion section. 

o Please explain how the hierarchical nature of DHS data would affect the overall prevalence

- Results

o Participants characteristics (table 1) presented in “data sources” should be briefly presented here

Answer: Thank you for your comment. We have displaced participants’ description to this section.

o Line 3: the 95% CI is very narrow, suggesting that clustering was not taken into account

Answer: We weighted data to account for the complex design, including clustering of the household survey.

o Line 12: (23%-25%): what is this range for?

Answer: The prevalence of malaria infection was low among children living with their mothers alone (23%) or with the two parents (25%) compared to those others (31%).

o Line 38: Box 1 and CHAID outputs show 5 significant predictors not 6 as mention here.

Answer: Thank you for your comment which has been implemented.

o Line 48: Missing residence

Answer: Thank you for your comment which has been implemented.

o Line 49: In Tanganyika, there is no predictor of malaria since the prevalence is very high (50%) for all children. Suggestion: No subsequent malaria infection predictor was identified in Tanganyika.

Answer: Thank you for the suggestion which has been implemented.

o Line 58: there’s no table 4

Answer: Thank you for the comment. Table 4 has been inserted. 

o Line 63: WHO classification: please specify classification of what?

Answer: Thank you for the comment. We have specified “WHO classification of malaria epidemiology”. 

o Clusters: Since these clusters are built by authors using the 26 final nodes of CHAID, I would expect malaria prevalence in each cluster to be presented as a range

Answer: We calculated the average for each cluster. Indicator within each cluster are from the 26 nodes.

o If the main purpose of creating the clusters is to summarize the population at risk (and protected) based on identified risk factors ( or protective factors), then cluster 3 is not informative as it includes the majority of the sample and all explanatory variables.

Answer: Thank you for your comment. We understand the problem you raised. These findings show the cluster where the majority of children belong and the complexity of their profile.

- Discussion

o Overall, the discussion is very limited and brief, to say the least. There are several points that could be discussed, but the authors shortly discussed 1 or 2 points only.

Answer: We have provided more discussion points.

o The recommendations provided are not supported by the findings of this paper

Answer: Recommendations are suggested by key findings. 

o Line 97: demographic as well

Answer: We revised the sentence

o Line 103 and 104: predictor of malaria infection

Answer: We revised the sentence

o Line 104: These are not “the main predictor” but identified predictors of malaria infection

Answer: We revised the sentence

o Line 105: The sentence starting with

Answer: We revised the sentence

---

## [Editor Report · Decision Letter 1]

23 Mar 2021

PONE-D-20-10152R1

Profiling malaria infection among under-five children in the Democratic Republic of Congo

PLOS ONE

Dear Dr. Emina,

Thank you for submitting your manuscript to PLOS ONE. After careful consideration, we feel that it has merit but does not fully meet PLOS ONE’s publication criteria as it currently stands. Therefore, we invite you to submit a revised version of the manuscript that addresses the points raised during the review process.

We note that you have addressed the majority of the reviewers' comments, but the manuscript as it stands requires some editing. I attach a version of the manuscript with suggested edits (most of which are English language edits).

A marked-up copy of your manuscript that highlights changes made to the original version. You should upload this as a separate file labeled 'Revised Manuscript with Track Changes'.An unmarked version of your revised paper without tracked changes. You should upload this as a separate file labeled 'Manuscript'.

We look forward to receiving your revised manuscript.

Kind regards,

Thomas A. Smith

Academic Editor

PLOS ONE
---

## [Author Response · Author response to Decision Letter 1]

7 Apr 2021

Dear Editor

We have edited the manuscript to meet Plos One standard. The ethics statement appears only in the Methods section.

---

## [Editor Report · Decision Letter 2]

12 Apr 2021

Profiling malaria infection among under-five children in the Democratic Republic of Congo

PONE-D-20-10152R2

Dear Dr. Emina,

We’re pleased to inform you that your manuscript has been judged scientifically suitable for publication and will be formally accepted for publication once it meets all outstanding technical requirements.

Kind regards,

Thomas A. Smith

Academic Editor

PLOS ONE
---

## [Editor Report · Acceptance letter]

21 Apr 2021

PONE-D-20-10152R2 

Profiling malaria infection among under-five children in the Democratic Republic of Congo 

Dear Dr. Emina:

I'm pleased to inform you that your manuscript has been deemed suitable for publication in PLOS ONE. Congratulations! Your manuscript is now with our production department. 

Kind regards, 

on behalf of

Prof. Thomas A. Smith 

Academic Editor

PLOS ONE